# Dynamics Adapted Imitation Learning

**Zixuan Liu**[1,2]*, **Liu Liu**[2]†, **Bingzhe Wu**[2], **Lanqing Li**[4], **Xueqian Wang**[1], **Bo Yuan**[3]†, **Peilin Zhao**[2]†
*zx-liu21@mails.tsinghua.edu.cn, leonliuliu@tencent.com, bingzhewu@tencent.com, lanqingli1993@gmail.com,*
*wang.xq@sz.tsinghua.edu.cn, boyuan@ieee.org, masonzhao@tencent.com*
[1] *Tsinghua University,* [2] *Tecent AI Lab,* [3] *Research Institute of Tsinghua University in Shenzhen,* [4] *Zhejiang Lab*

**Reviewed on OpenReview:** *https://openreview.net/forum?id=w36pqfaJ4t*

## Abstract

We consider Imitation Learning with dynamics variation between the expert demonstration (source domain) and the environment (target domain). Based on the popular framework of Adversarial Imitation Learning, we propose a novel algorithm – Dynamics Adapted Imitation Learning (DYNAIL), which incorporates the dynamics variation into the state-action occupancy measure matching as a regularization term. The dynamics variation is modeled by a pair of classifiers to distinguish between source dynamics and target dynamics. Theoretically, we provide an upper bound on the divergence between the learned policy and expert demonstrations in the *source domain*. Our error bound only depends on the expectation of the discrepancy between the source and target dynamics for the optimal policy in the *target domain*. The experiment evaluation validates that our method achieves superior results on high dimensional continuous control tasks, compared to existing imitation learning methods.

## 1 Introduction

For sequential decision making tasks, recent years have witnessed the success of Reinforcement Learning (RL) with carefully designed reward functions (e.g. Li et al., 2011; Mnih et al., 2015; Haarnoja et al., 2017; Thomas et al., 2017; Silver et al., 2018; Vinyals et al., 2019; Kendall et al., 2019; Bellemare et al., 2020; Sutton & Barto, 2018). However, it is challenging to define the reward signal in many scenarios, such as learning "socially acceptable" interactions (Fu et al., 2018; Qureshi et al., 2018; 2019), autonomous driving (Kuderer et al., 2015; Kiran et al., 2021), etc.

Recently, modern imitation learning (IL) methods (Ho & Ermon, 2016; Fu et al., 2018; Finn et al., 2016; Kostrikov et al., 2019; Ghasemipour et al., 2020; Orsini et al., 2021) propose to use adversarial approaches. More specifically, these Adversarial Imitation Learning (AIL) methods leverage the mechanism of GAN (Goodfellow et al., 2014): they alternate between training a policy whose trajectories cannot be distinguished from the expert's trajectories, and training a discriminator to distinguish between the generated policy and expert trajectories. It has been shown that using different reward functions (based on the discriminator above) in AIL corresponds to the different choices of divergences between the marginal state-action distribution of the expert and the policy. We refer the readers to Ghasemipour et al. (2020) and Orsini et al. (2021) for an in-depth discussion on this topic.

However, in real world scenarios, high quality expert demonstrations are often scarce. For expert demonstration efficiency considerations, limited expert demonstrations are applied to different environments with the same objective. Thus, there may exist dynamics variation between the expert demonstration and the environment. This issue has already been addressed in a line of research (Dulac-Arnold et al., 2020; Mankowitz et al., 2020; Eysenbach et al., 2021; Liu et al., 2022a; Fu et al., 2018; Qureshi et al., 2019; Kim et al., 2020; Liu et al., 2020; Kirk et al., 2021). Under the transfer learning framework (Pan & Yang, 2009), we are interested in the problem setting where the transition dynamics in the source domain (expert demonstration) and the

---

*This work is done when Zixuan Liu works as an intern in Tencent AI Lab.
†Corresponding authors.

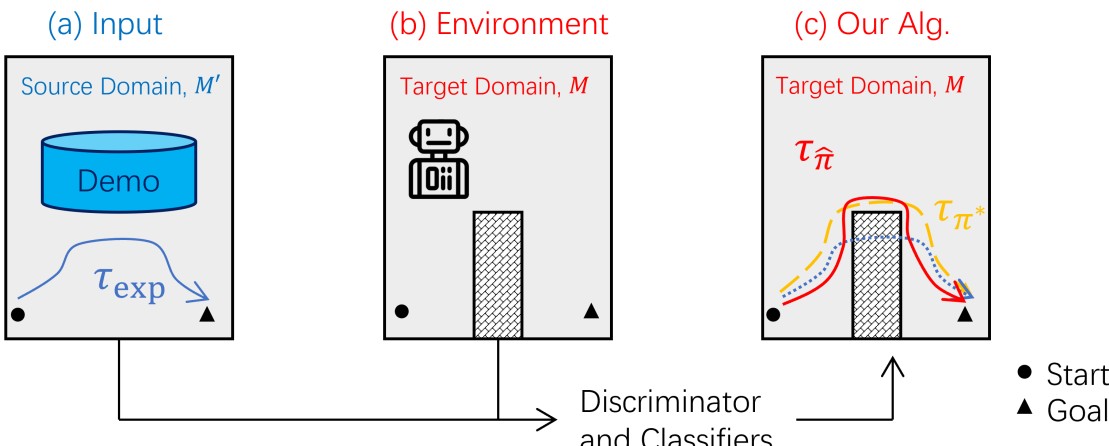

Figure 1: An illustration of Dynamics Adapted Imitation Learning problems *without* any reward signal. In (a), we are provided with the expert demonstrations $\mathcal{D}_{\text{demo}}$ with the underlying dynamics $M'$. $\tau_{\text{exp}}$ shows an expert's trajectory from the starting point (round) to the goal (triangle). In (b), an agent aims to make decisions in the target domain (environment) with the same observation space and (unknown) reward function yet different dynamics $M$. We inject the dynamics variation by adding an obstacle (shaded area). In (c), let $\pi^*$ denote the optimal policy in the target domain. The orange dashed line and red solid line are examples of the trajectories of $\pi^*$ and $\hat{\pi}$.

target domain (environment) are different. For example, in healthcare treatment, we have the same goal (reward function) for each patients, but they may react differently to treatments (transition dynamics) due to individual differences; in autonomous driving, it is common to encounter different road surfaces different from the expert demonstrations, which also change the transition dynamics. Hence, in this paper, we focus on the imitation learning problem under dynamics variation.

We illustrate the problem setting in Figure 1, where we use an obstacle to denote the dynamics variation. The source domain is for expert demonstrations and the target domain shares the same observation space and (unknown) reward function. However, the transition dynamics of the target domain $M$ is different from the source domain. It has been shown (e.g. Xu et al., 2021) that the policy value gap in Imitation Learning directly depends on the divergence between the state-action distribution of $\pi_{\text{exp}}$ and $\pi$ denoted by $\rho_{\text{exp}}$ and $\rho_{\pi}$ (please refer to Section 2 for formal definitions). In this paper, we directly use the divergence between $\rho_{\text{exp}}$ and $\rho_{\pi}$ as a *proxy for the policy value gap*, and we aim to find a policy $\hat{\pi}$ in the target domain, which has close performance closed to $\pi^*$.

## 1.1 Main contributions

- (**Algorithm**) We consider dynamics adaptation in imitation learning where the underlying transition dynamics of the expert demonstration (source domain) and the environment (target domain) are different. To tackle this issue, we propose a novel algorithm – Dynamics Adapted Imitation Learning (DYNAIL, Algorithm 1). The core ingredient of our algorithm is incorporating the dynamics variation into the state-action occupancy measure matching as a regularization term. The regularization for the dynamics discrepancy is modeled by a pair of classifiers to distinguish between source dynamics and target dynamics. Our algorithm builds upon the popular AIL framework, hence it's simple to implement, and computationally efficient.

- (**Theoretical guarantees**) We analyze our algorithm in terms of divergence minimization with a regularization on the discrepancy between the source and target dynamics. We establish a $\pi^*$-dependent upper bound on the suboptimality of our algorithm and has *no requirement* for all other possible policies, where $\pi^*$ is the optimal policy in the target domain. Although $\pi^*$ is unknown, our upper bounds only depends on the discrepancy between the source and target dynamics for $\pi^*$. Our

upper bounds explicitly account for the discrepancy between the source and target dynamics. It offers valuable insights into the relationship between the proposed method and the target domain's optimal policy (Section 4).

- (**Empirical validations**) We validate the effectiveness of DYNAIL on a variety of high-dimensional continuous control benchmarks with dynamics variations. Section 5 and Appendix show that our algorithm achieves superior results compared to state-of-the-art imitation learning methods.

## 1.2 Related work

**Domain adaptation in IL.** Most IL methods consider the unchanged transition dynamics (e.g. Pomerleau, 1988; Ho & Ermon, 2016; Brantley et al., 2019; Sasaki & Yamashina, 2021; Rajaraman et al., 2020; Liu et al., 2022b). Recently, there are different lines of research dealing with domain adaptation in imitation learning. Inverse RL methods – AIRL (Fu et al., 2018) and EAIRL (Qureshi et al., 2019) – are able to deal with dynamics variation in imitation learning settings with warm-up in source domain. Kim et al. (2020) and Raychaudhuri et al. (2021) achieve Markov decision process (MDP) alignment by interacting with both source and target domains. However, the source domain is inaccessible in most Imitation Learning settings since domains are much harder to store than demonstrations especially in real-world scenarios. Furthermore, (Kim et al., 2020) directly transferring expert policies from source domain to target domain requires online expert policy, which is an additional assumption to imitation learning settings. Liu et al. (2020) align state sequences to learn to focus on imitating states and the alignment of policy and its prior serves as a constraint in the policy update objective. The policy prior derives from the variational auto-encoder and (generally challenging) inverse dynamics model which are hard to train. Fickinger et al. (2021) utilize the Gromov-Wasserstein distance to compare different domains through the whole trajectories. However, sampling the whole trajectories instead of state-action joint distributions brings high variance. Chae et al. (2022) also deal with dynamics variation in Imitation Learning, however, it requires experiences from multiple source domains to complete the similar task in target domain. In the contrast, our method only needs demonstrations from one source domain to realize domain adaptation.

**Off-dynamics RL.** Different from IL, RL has direct access to the reward function. A recent survey (Kirk et al., 2021) provides a thorough introduction on the generalization of RL, which includes state, dynamics, observation or reward function variation. Most relevant to our work, Eysenbach et al. (2021) and Liu et al. (2022a) consider dynamics variation problems in RL. More specifically, Eysenbach et al. (2021) propose reward modification for optimizing the policy in the source domain to obtain the near-optimal policy in the target domain. Liu et al. (2022a) extend the idea of reward modification to the offline setting.

**Transfer RL.** Transfer RL is aimed at learning robust policy among several domains in RL settings. Unlike Off-dynamics RL, representation is a more popular topic than dynamics in this field. Some previous works focus on learning domain-agnostic representations (Zhang et al., 2020a;b; Tomar et al., 2021) to improve the generalization of representation, while other works (Tirinzoni et al., 2018; Huang et al., 2021) propose to make use of both domain-shared and domain-specific representations.

# 2 Background and Problem Setting

## 2.1 Markov Decision Process and Imitation Learning

We first introduce the background of MDP and IL *without* considering dynamics variation. An MDP $\langle \mathcal{S}, \mathcal{A}, r, M, d_0, \gamma \rangle$ consists of a state space $\mathcal{S}$, an action space $\mathcal{A}$, a reward function $r : \mathcal{S} \times \mathcal{A} \to \mathbb{R}$, an unknown transition dynamics $M : \mathcal{S} \times \mathcal{A} \to \Delta(\mathcal{S})$, an initial state distribution $d_0 \in \Delta(\mathcal{S})$, and a discounted factor $\gamma \in (0, 1)$. An agent following the policy $\pi(a|s)$ in an MDP with dynamics $M(s'|s, a)$ generates the trajectory $\tau_\pi = \{s_1, a_1, \cdots, s_T, a_T\}$, where $T$ is the termination step. We define the probability distribution $\rho_{\pi,t}^M(s, a) \in \Delta(\mathcal{S} \times \mathcal{A})$ to denote the marginal joint stationary distribution for state and action at time-step $t$. Then we define the discounted stationary state-action distribution $\rho_\pi^M(s, a) = (1 - \gamma) \sum_t \gamma^t \rho_{\pi,t}^M(s, a)$, and the discounted stationary state distribution $d_\pi^M(s) = \sum_{a \in \mathcal{A}} \rho_\pi^M(s, a)$. The goal of RL is to find a policy $\pi$ to maximize the expected cumulative reward $V_\pi = \mathbb{E}_{\tau \sim \pi} [\sum_t \gamma^t r(s_t, a_t)]$.

*Without* any reward signal, Imitation Learning algorithms (e.g. Osa et al., 2018) aim to obtain a policy to mimic the expert's behavior using a demonstration dataset $\mathcal{D}_{\text{demo}} = \{(s_i, a_i)\}_{i=1}^N$, where $N$ is the sample size of the dataset. In IL, we are interested in the policy value gap $V_{\text{exp}} - V_\pi$, where $V_{\text{exp}}$ is the value of the expert's policy. A classic type of IL methods is Behavior Cloning (Pomerleau, 1988), which uses supervised learning (e.g., maximum likelihood estimation) to learn the policy $\pi$.

Another line of research uses the idea of *Inverse RL* (Russell, 1998; Ng & Russell, 2000), which infers the expert's reward function from its demonstrations dataset, and then trains a policy to optimize this learned reward. Early works on Inverse RL (Abbeel & Ng, 2004) rely on matching feature expectations or moments between policies and experts. After that, Max Entropy Inverse RL (Ziebart et al., 2008; Ziebart, 2010) can handle the ambiguity problem of the policy optimization, where there are many optimal policies that can explain the expert demonstrations. Max Entropy Inverse RL addresses this issues by optimizing the maximum entropy objective to ensure that the solution is unique. More specifically, the following *generative model* is trained according to the trajectory $\tau$:

$$\max_\theta \mathbb{E}_{\tau \sim \mathcal{D}_{\text{demo}}}[\log p_\theta(\tau)], \text{ where } p_\theta(\tau) = \frac{1}{Z} p(s_0) \times \prod_{t=0}^T M(s_{t+1} \mid s_t, a_t) \exp(r_\theta(s_t, a_t)/\eta), \tag{1}$$

where $Z$ is the normalization for the parameterization of $p_\theta(\tau)$, and $\eta$ is the temperature parameter. The formulation Equation (1) connects the reward learning problem with maximum likelihood problem.

## 2.2 Adversarial Imitation Learning

Modern IL methods (Ho & Ermon, 2016; Fu et al., 2018; Finn et al., 2016; Ghasemipour et al., 2020; Orsini et al., 2021) use adversarial approaches for generative modeling by casting Equation (1) as a GAN (Goodfellow et al., 2014) optimization procedure, alternating between optimizing the discriminator and updating the policy. The discriminator deals with the binary classification by taking $(s, a)$ as the input, and assigning the expert trajectories positive labels and the generated samples negative labels, and then calculate the corresponding reward function. For example, AIRL (Fu et al., 2018) uses the following reward function:

$$r_\theta(s, a) = \log D_\theta(s, a) - \log(1 - D_\theta(s, a)). \tag{2}$$

It is shown that the minimization of the discriminator function is equivalent to the maximization of the likelihood function Equation (1). With the reward function Equation (2) in hand, we can use standard policy learning algorithms to update the policy.

It has been shown that, without dynamics variation, AIL methods are equivalent to minimizing the divergence between the $\rho_\pi(s, a)$ and $\rho_{\text{exp}}(s, a)$ (Ghasemipour et al., 2020; Xu et al., 2021), which often yield better performance with a few expert demonstrations compared to Behavior Cloning. Under the divergence minimization framework (Ghasemipour et al., 2020), AIRL is equivalent to

$$\widehat{\pi}^{\text{AIRL}} = \arg\min_\pi \mathbb{D}\left(\rho_\pi(s, a) \parallel \rho_{\text{exp}}(s, a)\right), \tag{3}$$

which is the so-called *reverse* KL divergence. Since the policy value gap in Imitation Learning directly depends on the divergence between $\rho_{\text{exp}}$ and $\rho_\pi$ (Xu et al., 2021), we directly use the divergence between $\rho_{\text{exp}}$ and $\rho_\pi$ as a *proxy for the policy value gap*.

## 2.3 Expert Demonstration Efficiency

Imitation learning, the method of learning from previously collected expert demonstrations, acquires near-optimal policy without the complex reward engineering in standard RL. However, collecting expert demonstration can be costly and the collected expert demonstrations are usually domain-specific with certain dynamics and policy information. Thus, vanilla imitation learning utilizing expensive expert demonstrations to complete the task in specific domains leads to low expert demonstration efficiency. A natural idea arises: expert demonstrations collected from a single source domain should be used for learning in different target domains, which can significantly improve expert demonstration efficiency (Liu et al., 2020; Fickinger et al., 2021; Chae et al., 2022).

### 2.4 Dynamics Adaptation in Imitation Learning

In this section, we consider Imitation Learning methods for dynamics adaptation. We use $M'(s' \mid s, a)$ to denote the transition dynamics in the source domain, and $M(s' \mid s, a)$ to denote the target domain. Directly applying vanilla Behavior Cloning or Inverse RL algorithms requires the same transition dynamics for the environment as the expert demonstrations (Fu et al., 2018), which limits their applicability. This is due to the fact that the learned rewards in vanilla Inverse RL are not transferable to environments featuring different dynamics.

There have been some attempts based on Inverse RL to obtain a reward function which is transferable and robust to dynamics variation (Fu et al., 2018; Qureshi et al., 2019). By minimizing the KL-divergence, these methods are able to deal with dynamics variation in the setting of imitation learning when the source environment is accessible for limited interactions to learn a robust reward function.

Other methods (e.g. Liu et al., 2020; Fickinger et al., 2021) use state distribution matching or trajectory matching rather than state-action distribution matching to realize dynamics adaptation. Liu et al. (2020) additionally train an inverse dynamics model and variational auto-encoder to derive a policy prior to regularize the state distribution matching. Fickinger et al. (2021) introduce the Gromov-Wasserstein distance into trajectory matching to find an isometric policy.

We introduce the following assumption on the transition dynamics for our problem setting.

**Assumption 1.** *If the transition probability in the target domain is positive, then the transition probability in the source domain is positive, i.e., $M(s' \mid s, a) > 0 \Rightarrow M'(s' \mid s, a) > 0, \ \forall s, s' \in \mathcal{S}, a \in \mathcal{A}$.*

This assumption is modest and common in previous literature (e.g. Koller & Friedman, 2009; Eysenbach et al., 2021). If it does not hold, the trajectory induced by the optimal policy in the target domain $\pi^*$ might incur invalid behaviors in the source domain.

## 3 Our Algorithm

To better identify the role of the transition dynamics in the distribution of the trajectories, we directly compare the distribution over the trajectories. In the source domain with transition dynamics $M'$, the goal of Max Entropy Inverse RL (Ziebart et al., 2008) is to maximize the likelihood given the expert demonstrations data $\mathcal{D}_{\text{demo}}$ by the following parameterization.

$$\max_\theta \mathbb{E}_{\tau \sim \mathcal{D}_{\text{demo}}}[\log p_\theta(\tau; \text{source})], \tag{4}$$

$$p_\theta(\tau; \text{source}) = \frac{1}{Z} p(s_0) \times \prod_t M'(s_{t+1} \mid s_t, a_t) \exp(r_\theta(s_t, a_t)/\eta), \tag{5}$$

where $Z$ is the normalization of $p_\theta(\tau; \text{source})$, and $\eta$ is the temperature parameter.

On the other hand, for the target domain with transition dynamics $M$, the policy $\pi_\theta(a_t \mid s_t)$ yields the distribution over trajectory $\tau$:

$$p_\pi(\tau; \text{target}) = p(s_0) \prod_t M(s_{t+1} \mid s_t, a_t) \pi(a_t \mid s_t). \tag{6}$$

### 3.1 Divergence minimization with dynamics shift

Based on the divergence minimization framework for imitation learning (Ho & Ermon, 2016; Finn et al., 2016; Fu et al., 2018; Ghasemipour et al., 2020; Eysenbach et al., 2021), our motivation is to learn a policy $\pi$ running under the target domain dynamics whose behavior has high likelihood based on the expert demonstrations. Hence we minimize the reverse KL divergence[1]

$$\min_\pi \mathbb{D}\left(p_\pi(\tau; \text{target}) \parallel p_\theta(\tau; \text{source})\right). \tag{7}$$

---

[1]The KL divergence has been widely used in the RL literature, and we leave the discussion to the Appendix A.

Expanding Equation (7) yields the policy objective Equation (8)

$$\max_\pi \mathbb{E}_{\tau \sim \pi, M} \left[ \sum_t r_\theta(s_t, a_t) - \eta \log \pi(a_t \mid s_t) + \underbrace{\eta \Phi(s_t, a_t, s_{t+1})}_{\text{Discrepancy Regularization}} \right], \qquad (8)$$

where the discrepancy regularization equals

$$\Phi(s_t, a_t, s_{t+1}) = \log M'(s_{t+1} \mid s_t, a_t) - \log M(s_{t+1} \mid s_t, a_t).$$

Note that we neglect the normalization term in Equation (8). Following the AIL framework introduced in Section 2.2, the reward function term $r_\theta$ can be obtained from generating the policy in the environment and training a discriminator. The $\log \pi(a_t \mid s_t)$ term is considered to be the entropy regularizer. These two terms are actually equivalent to the vanilla AIRL (Fu et al., 2018).

Importantly, the regularization term $\Phi(s_t, a_t, s_{t+1})$ represents the dynamics shift from $M'$ to $M$ by calculating $\log M'(s_{t+1} \mid s_t, a_t) - \log M(s_{t+1} \mid s_t, a_t)$. When summed over the trajectory induced by $\pi$ under the dynamics $M$, this is equivalent to a regularization term

$$\mathbb{E}_{(s,a) \sim \rho_\pi^M} \left[ -\mathbb{D} \left( M(\cdot \mid s, a) \parallel M'(\cdot \mid s, a) \right) \right]$$

in the reward maximizing objective. When the source and target dynamics are equal, the regularization term $\Phi$ is zero and our method reduces to the vanilla AIL methods (Ho & Ermon, 2016; Fu et al., 2018).

This discrepancy between the source and target dynamics has been previously applied to dynamics adaptation in RL literature (e.g., online RL (Eysenbach et al., 2021) and offline RL(Liu et al., 2022a)). We provide a novel – and to the best of our knowledge, the first algorithm to incorporate the dynamics discrepancy in the state-action occupancy measure matching for Imitation Learning problems. In the next subsection, we propose practical methods (see Equation (11) and Equation (12)) to estimate $\Phi(s, a, s')$.

## 3.2 Dynamics Adapted Imitation Learning

To maximize the likelihood function over the trajectories of the experts Equation (4), we use adversarial approaches for generative modeling by casting Equation (1) as a GAN (Goodfellow et al., 2014) optimization. As introduced in Section 2, we leverage the reward function formulation $r = \log(\frac{D}{1-D})$, which is equivalent to distribution matching under KL divergence as in Equation (3). We summarize our algorithm Dynamics Adapted Imitation Learning (DYNAIL) in Algorithm 1. The overall procedure of our algorithm follows from the GAN (Goodfellow et al., 2014) which alternates between optimizing the discriminator/classifiers and updating the policy.

**Training the discriminator.** The discriminator $D_\theta$ is designed to distinguish expert policy and current running $\pi$. The training of the discriminator $D_\theta$ follows from

$$\min_{D_\theta} -\mathbb{E}_{\tau \sim \mathcal{D}_{\text{demo}}} \left[ \log D_\theta(s_t, a_t) \right] - \mathbb{E}_{\tau \sim \pi, M} \left[ \log(1 - D_\theta(s_t, a_t)) \right], \qquad (10)$$

where we train the discriminator, $D_\theta = \frac{\exp(r_\theta/\eta - \Phi)}{\log \pi + \exp(r_\theta/\eta - \Phi)}$, by labelling the expert demonstrations $\mathcal{D}_{\text{demo}}$ from source domain as positive. The negative samples are the trajectory from running policy $\pi$ in the target domain under the dynamics $M$. As justified in Fu et al. (2018), training on $D_\theta$ results in optimizing the likelihood function in Equation (4).

**Training the classifiers.** The discrepancy regularization term is obtained by training two classifiers,

$$\Phi(s_t, a_t, s_{t+1}) = \log M'(s_{t+1} \mid s_t, a_t) - \log M(s_{t+1} \mid s_t, a_t)$$
$$\overset{(i)}{=} \log \frac{\Pr(\text{source}|s, a, s')}{\Pr(\text{target}|s, a, s')} - \log \frac{\Pr(\text{source}|s, a)}{\Pr(\text{target}|s, a)},$$

where $(i)$ follows from the fact that $M'(s'|s, a) = \Pr(s'|s, a, \text{source}) = \frac{\Pr(\text{source}|s, a, s') \Pr(s, a, s')}{\Pr(\text{source}|s, a) \Pr(s, a)}$, and $M(s'|s, a) = \Pr(s'|s, a, \text{target}) = \frac{\Pr(\text{target}|s, a, s') \Pr(s, a, s')}{\Pr(\text{target}|s, a) \Pr(s, a)}$.

---

**Algorithm 1** Dynamics Adapted Imitation Learning (DYNAIL)

---

1: **Input:** Expert demonstrations $\mathcal{D}_{\text{demo}}$ from the source domain, hyperparameter $\eta$.
2: **Output:** Policy $\widehat{\pi}$ in the target domain with *different dynamics*.

---

3: Randomly initialize $\pi$.
4: **for** $t = 0$ to $T - 1$ **do**
5:      Sample trajectories $\tau = \{(s_t, a_t, s_t')\}_{t=1}^{T}$ by running $\pi$ in the target domain's environment.
6:      Train the discriminator $D_\theta$: the expert demonstrations $\mathcal{D}_{\text{demo}}$ from source domain are positive and the trajectory from policy $\pi$ are negative via Equation (10).
7:      Train the classifier $q_{\text{sa}}$ and $q_{\text{sas}}$ with cross-entropy loss via Equation (11) and Equation (12).
8:      Update the reward function $\widetilde{r}(s, a, s')$ via the following equation,

$$\widetilde{r}(s, a, s') = \log(D_\theta) - \log(1 - D_\theta) + \eta \left( \log \frac{q_{\text{sas}}}{1 - q_{\text{sas}}} - \log \frac{q_{\text{sa}}}{1 - q_{\text{sa}}} \right). \tag{9}$$

9:      Update the policy $\pi$ with respect to $\{(s_t, a_t, s_t', \widetilde{r}_t)\}_{t=1}^{T}$ using standard RL algorithm.
10: **end for**
11: **Return:** Policy $\pi$.

---

Hence we parameterize $\Pr(\text{source}|s, a, s')$ and $\Pr(\text{source}|s, a)$ with the two classifiers $q_{\text{sas}}(s, a, s')$ and $q_{\text{sa}}(s, a)$ respectively. We optimize the cross-entropy loss for the binary classification by labelling the trajectories $\mathcal{D}_{\text{demo}}$ from source domain as positive. The negative samples are the trajectory from running policy $\pi$ in the target domain under the dynamics $M$.

$$\min_{q_{\text{sas}}} - \mathbb{E}_{\tau \sim \mathcal{D}_{\text{demo}}} \left[ \log q_{\text{sas}}(s, a, s') \right] - \mathbb{E}_{\tau \sim \pi, M} \left[ \log(1 - q_{\text{sas}}(s, a, s')) \right], \tag{11}$$

$$\min_{q_{\text{sa}}} - \mathbb{E}_{\tau \sim \mathcal{D}_{\text{demo}}} \left[ \log q_{\text{sa}}(s, a) \right] - \mathbb{E}_{\tau \sim \pi, M} \left[ \log(1 - q_{\text{sa}}(s, a)) \right]. \tag{12}$$

Hence the discrepancy regularization $\Phi(s_t, a_t, s_{t+1}) = \log \frac{q_{\text{sas}}(s_t, a_t, s_{t+1})}{1 - q_{\text{sas}}(s_t, a_t, s_{t+1})} - \log \frac{q_{\text{sa}}(s_t, a_t)}{1 - q_{\text{sa}}(s_t, a_t)}$. Compared to the vanilla AIRL (Fu et al., 2018), we only incur computational overhead by training two additional classifiers. We note that we only learn the classifiers rather than the transition models. Hence our method is suitable to handle high-dimensional tasks, which is validated on a 111-dimensional ant environment in Section 5.

**Updating the policy.** With Equation (9) in hand, we can update the reward function for the collected trajectory – $\{(s_t, a_t, s_t', \widetilde{r}_t)\}_{t=1}^{T}$. We note that $\eta$ is the hyperparameter for the regularization term in the reward function. Besides the entropy term which is considered for policy regularization, optimizing the policy objective $\pi$ w.r.t. Equation (9) shares the same structure as Equation (8).

## 4 Theoretical Analysis

Let $\pi_{\text{exp}}$ denote the expert policy in the *source domain*, and $\rho_{\text{exp}}(s, a)$ denote its corresponding distribution of $(s, a)$. As introduced in Equation (3), AIL methods can be interpreted as minimizing the divergence between $\rho_\pi(s, a)$ and $\rho_{\text{exp}}(s, a)$ (Ho & Ermon, 2016; Ghasemipour et al., 2020; Xu et al., 2021), and this serves as a proxy for the policy value gap. Our theoretical analysis shows that by imposing the discrepancy regularization on the objective function, our error bound for the KL divergence compared to $\pi^*$ only depends on the discrepancy between the source and target dynamics for the expert policy in the target domain.

To distinguish the expert demonstrations and the generated trajectory of the policy $\pi$, the optimal discriminator for Equation (10) (line 6 of Algorithm 1) is achieved[2] at

$$D^*(s, a) = \frac{\rho_{\text{exp}}(s, a)}{\rho_{\text{exp}}(s, a) + \rho_\pi^M(s, a)}.$$

---

[2] We consider the asymptotic results and assume that the optimal $D_\theta$ can be achieved.

We note that a simple proof was included in Goodfellow et al. (2014). We plug in this $D^*(s, a)$ to the reward function $\log(\frac{D}{1-D})$, and clip the logits of the discriminator to the range $[-C, C]$ ($C$ is a universal constant such as 10). The Equation (9) in Algorithm 1 with $D^*$ is equivalent to the following optimization problem with $\eta\Phi$ as a regularization term

$$\max_\pi \mathbb{E}_{\tau \sim \pi, M} \left[ \sum_t \log \rho_{\exp}(s_t, a_t) - \log \rho_\pi^M(s_t, a_t) + \eta\Phi(s_t, a_t, s_{t+1}) \right], \tag{13}$$

where the temperature $\eta > 0$ is a tuning regularization hyperparameter to be chosen. According to KKT condition, there exists an $\varepsilon > 0$, such that regularized formulation Equation (13) is equivalent to the constrained formulation (The equivalence is straightforward and we leave the details to the Appendix A) Equation (14), where the deviation in KL divergence with different dynamics for any policy $\pi \in \Pi_\varepsilon$ is controlled by some $\varepsilon$.

$$\max_\pi \mathbb{E}_{(s,a) \sim \rho_\pi^M} \left[ \log \rho_{\exp}(s, a) - \log \rho_\pi^M(s, a) \right], \text{subject to } \pi \in \Pi_\varepsilon, \tag{14}$$

$$\text{where } \Pi_\varepsilon = \left\{ \pi \mid \mathbb{E}_{(s,a) \sim \rho_\pi^M} \left[ \mathbb{D} \left( M(\cdot \mid s, a) \parallel M'(\cdot \mid s, a) \right) \right] \le \varepsilon \right\}. \tag{15}$$

By leveraging the regularization/constraint to the reward maximizing optimization, we show that, surprisingly, our theoretical guarantees only depend on the smallest positive $\varepsilon$ of Equation (15) for the optimal policy in the target domain.

**Definition 4.1.** *Let $\pi^*$ be the optimal policy in the target domain, and $\varepsilon^*$ be the smallest positive solution of Equation (15) for $\pi^*$, i.e.,*

$$\mathbb{E}_{(s,a) \sim \rho_{\pi^*}^M} \left[ \mathbb{D} \left( M(\cdot \mid s, a) \parallel M'(\cdot \mid s, a) \right) \right] \le \varepsilon^*.$$

We note that the $\varepsilon^*$ in Definition 4.1 can always be achieved, since $\eta$ is a tuning hyperparameter. And for a fixed pair of $M$ and $M'$, a smaller hyperparameter $\eta$ corresponds to a larger $\varepsilon$. Then there always exists a pair of $\eta$ and $\varepsilon$ such that Definition 4.1 is satisfied for $\pi^*$. Although the $\pi^*$ is unknown, this critical quantity $\varepsilon^*$ only depends on the discrepancy between the source and target dynamics for $\pi^*$, rather than any other policy candidates.

Our Theorem 4.1 bounds the KL divergence between $\rho_\pi^{M'}(s, a)$ and $\rho_{\exp}(s, a)$. We focus on the term $\rho_\pi^{M'}(s, a)$, which denotes the state-action distribution while executing the policy $\hat\pi$ in the *source domain* with the transition dynamics $M'$. Since $\rho_{\exp}$ is the provided expert demonstration, the upper bound for the quantity $\mathbb{D}_{\mathrm{KL}}(\rho_\pi^{M'} \parallel \rho_{\exp})$ directly relates to the sub-optimality of the value function, which is a natural metric as introduced in Section 2.2. Compared to executing $\pi^*$ (the optimal policy in the target domain) in the source domain, the KL divergence with the optimal $\rho_{\exp}(s, a)$ only incurs an additive error of $O(\frac{\gamma}{1-\gamma}\sqrt{\varepsilon^*})$.

> **Theorem 4.1.** *Let $\rho_{\exp}(s, a)$ denote the distribution of $(s, a)$ for the expert policy in the source domain, and $\pi^*$ be the optimal policy in the target domain. Under Assumption 1 and Definition 4.1, we have*
>
> $$\mathbb{D} \left( \rho_\pi^{M'} \parallel \rho_{\exp} \right) \le \mathbb{D} \left( \rho_{\pi^*}^{M'} \parallel \rho_{\exp} \right) + O(\frac{\gamma}{1-\gamma}\sqrt{\varepsilon^*}). \tag{16}$$

**Remark 4.1.** *The theoretical analysis is restricted to the source domain. Due to the nature of Imitation Learning, we provide the upper bound for the quantity $\mathbb{D}_{\mathrm{KL}}(\rho_\pi^{M'} \parallel \rho_{\exp})$. If the reward signal is provided, we can extend the bounds to the target domain. The proof is deferred to the Appendix A. In Equation (16), we provide theoretical upper bounds by the "$\pi^*$-dependent" quantity $\varepsilon^*$, and has no requirement for all other possible policies. Following the previous statistics literature (Donoho & Johnstone, 1994; Wainwright, 2019), we refer to such bounds as the oracle property – the error upper bounds can automatically "adapt" to the dynamics discrepancy of the trajectory induced by $\pi^*$. Although the optimal policy $\pi^*$ in the target domain is unknown, our upper bound identifies the discrepancy of KL divergence for $\pi^*$ via the "oracle" property. Furthermore, the linear dependency on $\frac{1}{1-\gamma}$ (which is the effective horizon for the discounted MDP) matches the best known results (Rajaraman et al., 2020; Xu et al., 2021) for IL methods. When the dynamics in the source domain and target domain are the same, i.e., $\varepsilon \to 0$, and we recover $\hat\pi = \pi^*$, which is also the expert policy in the source domain.*

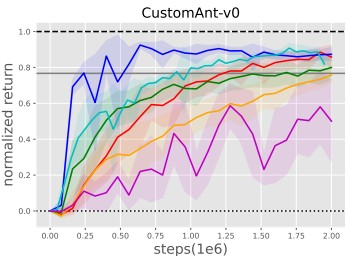 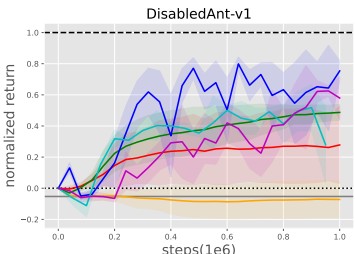 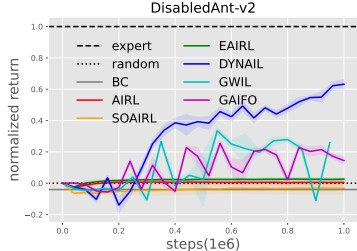

(a) The normalized return vs. the iterations on different target domains. The dashed line stands for the performance of $\pi^*$, which is the optimal results in each target domain. BC is a straight line for no interaction with the target domain. The shaded area stands for one standard deviation for 20 trials. In the left plot, the target domain is the same as the source domain, and all the methods have good convergence properties. When there exists dynamics variations (middle and right plots) DYNAIL has superior performance compared to AIRL, SOAIRL (the shorthand for state-only AIRL), EAIRL, GAIFO, GWIL and BC. The performance gap is widening as $n_{\mathrm{crippledlegs}}$ grows (from left to right).

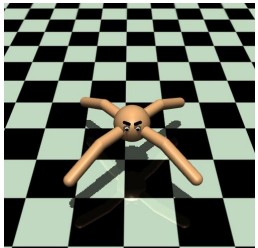 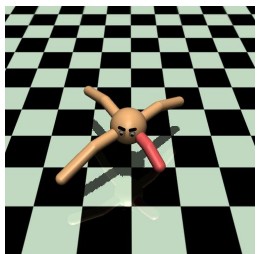 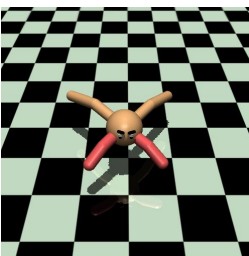

(b) The snapshot in the corresponding target environments. Different levels of dynamics variations are represented by different $n_{\mathrm{crippledlegs}}$ in CustomAnt-v0, DisabledAnt-v1 and DisabledAnt-v2 (from left to right).

Figure 2: Performance on different target domains for ant environment. For all the experiments, we use the same pre-collected 40 expert trajectories on source domain (CustomAnt-v0) as expert demonstrations.

## 5 Experiments

In this paper, we propose an imitation learning method which deals with the dynamics variation between demonstrations and interactions. Our experiments aim to investigate the following questions: (1) Does our method outperform prior work in dynamics adaptation? (2) What degree of dynamics shift can our method address? We briefly present the performance of our algorithm in this section, and defer the ablation study to the Appendix B.

**Experimental setup.** We compare our algorithm DYNAIL with a number of baselines introduced in Section 2.4. A natural baseline is to use Behavior Cloning (BC) on the expert demonstrations, and then directly apply it to the target domain without the consideration of dynamics variation. We investigate AIRL (Fu et al., 2018), its variant State-only AIRL (SOAIRL), Empowered AIRL (EAIRL) (Qureshi et al., 2019) and GAIFO (Torabi et al., 2018). We also evaluate our algorithm by comparing its performance against the state-of-the-art baseline GWIL (Fickinger et al., 2021).

Since these baselines and our algorithm share the similar training procedure, we use the same net architecture. We use PPO (Schulman et al., 2017) for the generator in AIL framework except for humanoid task where we use SAC (Haarnoja et al., 2018) for the generator, to optimize the policy and use 10 parallel environments to collect transitions on target domains. The discriminator $D_\theta$, classifiers $q_{\mathrm{sa}}$ and $q_{\mathrm{sas}}$ have the same structure of hidden layers, 2 layers of 256 units each, and a normalized input layer. We use ReLU as activation after each hidden layer. In all experiments, discounting factor is considered as 0.99. A key hyperparameter for our method is $\eta$, which serves as a tuning regularization. and we defer the full ablation study on $\eta$ to the Appendix B.

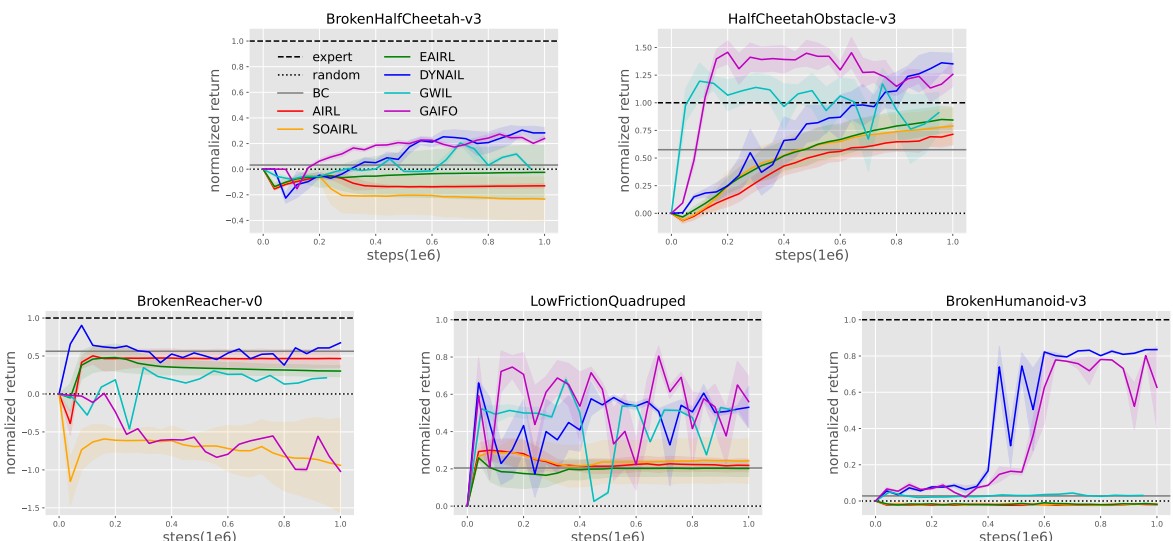

Figure 3: The normalized return vs. the iterations on different target domains for continuous control tasks. The baselines are the same with Figure 2, hence the descriptions are omitted. For all the experiments, we use the same pre-collected 40 expert trajectories on source domain (HalfCheetah-v3, Reacher-v0, Quadruped and Humanoid-v3) as expert demonstrations, and the snapshots for different environments are leaved to Figure 5 in the Appendix.

For all the experiments, the expert demonstrations are collected by using RL algorithms in Stable Baselines3 (Raffin et al., 2019) by interacting with the *source domains* without access to *target domains*. We use 40 trajectories collected by expert as demonstrations. For all the environments used in our experiments have fixed horizon, which is the maximum length of episode in the environment.

## 5.1 Different levels of dynamics variation

For the environment setup, we follow the simulator previously used in Qureshi et al. (2019), where we use a custom ant as source domain and several differently disabled ant as target domains.

**Custom ant.** Custom ant is basically the same as ant from OpenAI Gym (Brockman et al., 2016) except for joint gear ratios. With lower joint gear ratios, the robot flips less often and the agent learns fast. We refer this environment as CustomAnt-v0. The dimension of observation space is 111 and that of action space is 8.

**Differently disabled ant.** Disabled ant differs custom ant in crippled legs whose impact on the dynamics of robot is bigger than crippled joint in broken ant domain. The degree of disability depends on the number of crippled legs, $n_{\text{crippledlegs}} = 1, 2$, which corresponds to DisabledAnt-v1 and DisabledAnt-v2.

We present the experimental results in Figure 2. The experts represented by dashed lines in the plots are RL agents trained with reward in the target domain. The shaded area of all the curves stands for standard deviation for 20 trials. BC does not require interaction with the target domain, hence its performance are straight lines in the plots. In the left plot, the target domain CustomAnt-v0 is the same as the source domain, and all the methods have good convergence properties. With the increase of $n_{\text{crippledlegs}}$, we observe that the performance of baselines decreases, yet our method still outperforms the baselines in different target domains. This demonstrates the effectiveness of our method for imitation learning in the target domain with dynamics variation.

## 5.2 Different continuous control tasks

We extend our experiments to four continuous control tasks with crippled bodies and obstacles, and a real-world scenario[3]. The target environments derived from standard environments are widely studied in previous works (Eysenbach et al., 2021; Dulac-Arnold et al., 2020). Due to space limit, we defer the detailed setup of the target environments to the Appendix B.

We present the experimental results in Figure 3. In all the experiments with dynamics variations, our algorithm DYNAIL achieves superior performance compared to the baselines. We note that in HalfCheetahObstacle-v3, our method even obtains higher reward than the "optimal" policy in the target domain which is trained via standard RL algorithm in the target domain. This is due to that it is easier to run backwards in HalfCheetahObstacle-v3, the expert tends to run backwards and is prevented by the obstacle with limited reward. However, our algorithm DYNAIL learns to run forwards and gets more reward with the assistance of demonstrations provided by expert running forwards in the source domain. This phenomenon for the environment is also observed in Eysenbach et al. (2021).

## 6 Conclusion and future work

We've proposed a novel algorithm to incorporate dynamics variation into imitation learning and learns a superior near-optimal policy to the baselines. Compared with prior work (Liu et al., 2020; Kim et al., 2020; Raychaudhuri et al., 2021; Fickinger et al., 2021), it is achieved without access to source domain or any assistance of the online expert and complex dynamics model. There are several avenues for the future work: since our work is actually a Meta-Algorithm designed for dynamics variation, which builds upon Adversarial Imitation Learning. We can leverage previous work on AIL to improve the sample complexity (Kostrikov et al., 2019). Any improvement in the time/sample complexity of AIL methods would benefit from the robustness to dynamics variation by our algorithm. Also, we can leverage a Real world RL control suite (Dulac-Arnold et al., 2020), Grid2Op (Donnot, 2020) and CityLearn (Vazquez-Canteli et al., 2020) for the experiments, it would be of interest to apply our algorithm with real data, such as autonomous driving, girdpower operation and demand response.

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

# A    Proofs

## A.1    Facts about KL divergence

We introduce Assumption 1 on the transition dynamics, which assumes that if $M(s' \mid s, a) > 0$, then $M'(s' \mid s, a) > 0$, $\forall s, s' \in \mathcal{S}, a \in \mathcal{A}$. If it does not hold, then the trajectory induced by the optimal policy in the target domain $\pi^*$ might involve invalid behaviors in the source domain. The reverse of Assumption 1 is not necessary to be true.

The usage of KL divergence in Imitation Learning, Reinforcement Learning and Control is ubiquitous (Fu et al., 2018; Levine, 2018; Ghasemipour et al., 2020). Similar assumptions on transition dynamics are also widely used in Koller & Friedman (2009) and Eysenbach et al. (2021). We note the Assumption 1 is also consistent with the definition of KL divergence in Equation (7).

$$\min_{\pi} \mathbb{D}\left(p_\pi(\tau; \text{target}) \parallel p_\theta(\tau; \text{source})\right).$$

However, such assumptions may not hold for simulation environments with deterministic transition dynamics such as MuJoCo, due to mathematical definition. Hence, it is found important for the practical implementation to use the combination of clipped discriminator logits, and tuning the gradient penalty for effective training of Adversarial Imitation Learning methods (Ghasemipour et al., 2020; Orsini et al., 2021).

## A.2    Proof of Theorem 4.1

Based on the constrained optimization formulation Equation (14), we first build a uniform bound (Lemma A.1) for the deviation of all policies in the constraint set $\Pi_\varepsilon$.

**Lemma A.1** (Uniform Bound). *Under Assumption 1, for all the policy in the constraint set $\Pi_\varepsilon$, we have*

$$|\mathbb{E}_{\rho_\pi^M}\left[\log\rho_{\exp} - \log\rho_\pi^M\right] - \mathbb{E}_{\rho_\pi^{M'}}\left[\log\rho_{\exp} - \log\rho_\pi^{M'}\right]| = O(\frac{\gamma}{1-\gamma}\sqrt{\varepsilon}), \forall \pi \in \Pi_\varepsilon, \tag{17}$$

*where $\Pi_\varepsilon = \left\{\pi \mid \mathbb{E}_{(s,a)\sim\rho_\pi^M}\left[\mathbb{D}\left(M(\cdot \mid s, a) \parallel M'(\cdot \mid s, a)\right)\right] \leq \varepsilon\right\}$.*

*Proof of Lemma A.1.* For the LHS of Equation (17), we consider the following function

$$\mathbb{E}_{(s,a)\sim\rho_\pi}\left[\log\frac{\rho_{\exp}(s,a)}{\rho_\pi(s,a)}\right] = -\mathbb{D}\left(\rho_\pi \parallel \rho_{\exp}\right),$$

which is actually the KL divergence function between $\rho_\pi$ and $\rho_{\exp}$. The KL divergence is Lipschitz continuous with respect to $\rho_\pi(s, a)$ in $\ell_1$ norm. This is due to the constant $C$ for the upper and lower bound on the logits of the discriminator.

We then present the perturbation bound on $\ell_1$ norm for the stationary function of state $d_\pi^M$. The techniques are from the perturbation theory (Schulman et al., 2015; Xu et al., 2021). Let $P_\pi^M(s' \mid s) = \sum_{a\in\mathcal{A}} M(s' \mid s, a)\pi(a|s)$ as the *state transition function* for a given policy $\pi$. Then the stationary distribution of $d$ can be written as

$$d_\pi^M = (1-\gamma)\sum_{t=0}^{\infty}\gamma^t \Pr(s_t = s; \pi, M, d_0)$$
$$= (1-\gamma)(I - \gamma P_\pi^M)^{-1}d_0$$

where $d_0$ is the initial state distribution.

Then we calculate the difference in the distribution

$$d_\pi^{M'} - d_\pi^M = \gamma(I - \gamma P_\pi^{M'})^{-1}(P_\pi^{M'} - P_\pi^M)d_\pi^M$$

Based on the bound in Lemma 7 of Xu et al. (2021), we have

$$\|d_\pi^{M'} - d_\pi^M\|_1 \leq \gamma\|(I - \gamma P^{M'})^{-1}\|_1\|(P^{M'} - P^M)d_\pi^M\|_1 \tag{18}$$

$$\overset{(i)}{\leq} \frac{\gamma}{1-\gamma}\|(P^{M'} - P^M)d_\pi^M\|_1 \tag{19}$$

$$\overset{(ii)}{\leq} \frac{\gamma}{1-\gamma}\mathbb{E}_{(s,a)\sim\rho_\pi^M}\left[|M(\cdot \mid s,a) - M'(\cdot \mid s,a)|\right] \tag{20}$$

$$\overset{(iii)}{\leq} \frac{\sqrt{2}\gamma}{1-\gamma}\sqrt{\varepsilon}, \tag{21}$$

where (i) follows from the fact $\left\|(I - \gamma P^{M'})^{-1}\right\|_1 \leq \sum_{t=0}^{\infty}\gamma^t = \frac{1}{1-\gamma}$, (ii) follows from the definition of $P_\pi^M(s' \mid s)$, and (iii) follows from the Pinsker's inequality for bounding the $\ell_1$ norm via KL divergence, and the definition for $\varepsilon$. Similar techniques are also used in Schulman et al. (2015) and Xu et al. (2021).

The last step is bounding the $\left\|\rho_\pi^{M'}(s,a) - \rho_\pi^M(s,a)\right\|_1$ by $\left\|d_\pi^{M'} - d_\pi^M\right\|_1$ in Equation (20),

$$\|\rho_\pi^{M'}(s,a) - \rho_\pi^M(s,a)\|_1 \overset{(i)}{\leq} \|d_\pi^{M'} - d_\pi^M\|_1$$

$$\leq \frac{\sqrt{2}\gamma}{1-\gamma}\sqrt{\varepsilon},$$

where (i) follows from the definition of $\rho_\pi(s,a)$ and $d_\pi$.

Putting together the pieces, we have

$$|\mathbb{E}_{\rho_\pi^M}\left[\log\rho_{\exp} - \log\rho_\pi^M\right] - \mathbb{E}_{\rho_\pi^{M'}}\left[\log\rho_{\exp} - \log\rho_\pi^{M'}\right]| = O(\frac{\gamma}{1-\gamma}\sqrt{\varepsilon})$$

for all the policies in the constraint set $\Pi_\varepsilon$. $\qquad\square$

**Theorem A.1** (Theorem 4.1). *Let $\rho_{\exp}(s,a)$ denote the distribution of $(s,a)$ for the expert policy in the source domain, and $\pi^*$ be the optimal policy in the target domain. Under Assumption 1 and Definition 4.1, we have*

$$\mathbb{D}\left(\rho_\pi^{M'} \parallel \rho_{\exp}\right) \leq \mathbb{D}\left(\rho_{\pi^*}^{M'} \parallel \rho_{\exp}\right) + O(\frac{\gamma}{1-\gamma}\sqrt{\varepsilon^*}). \tag{22}$$

*Proof of Theorem 4.1.* To distinguish the expert demonstrations and the generated trajectory of the policy $\pi$, the optimal discriminator for Equation (10) (line 6 of Algorithm 1) is achieved at

$$D^*(s,a) = \frac{\rho_{\exp}(s,a)}{\rho_{\exp}(s,a) + \rho_\pi^M(s,a)}. \tag{23}$$

The simple proof was included in Goodfellow et al. (2014), and the logits of the discriminator is clipped to be within the range, for example, $[-10, 10]$ (Ghasemipour et al., 2020).

Plugging in the Equation (23), and we are solving the RL problem with the discrepancy regularizer

$$\max_\pi \mathbb{E}_{\tau\sim\pi,M}\left[\sum_t \log\rho_{\exp}(s_t, a_t) - \log\rho_\pi^M(s_t, a_t) + \eta\Phi(s_t, a_t, s_{t+1})\right], \tag{24}$$

Since we have $\Phi(s_t, a_t, s_{t+1}) = \log M'(s_{t+1} \mid s_t, a_t) - \log M(s_{t+1} \mid s_t, a_t)$, the regularization term is actually equivalent to the constraint set

$$\Pi_\varepsilon = \left\{\pi \mid \mathbb{E}_{(s,a)\sim\rho_\pi^M}\left[\mathbb{D}\left(M(\cdot \mid s,a) \parallel M'(\cdot \mid s,a)\right)\right] \leq \varepsilon\right\}$$

Hence we have the following optimization in the domain of $\rho$.

$$\max_{\pi} \mathbb{E}_{(s,a)\sim\rho_{\pi}^{M}} \left[ \log \rho_{\exp}(s,a) - \log \rho_{\pi}^{M}(s,a) \right], \text{subject to } \pi \in \Pi_{\varepsilon}. \tag{25}$$

The next step is analyzing the objective function of Equation (25) – based on the optimality condition of $\widehat{\pi}$ in Algorithm 1 and the feasibility of $\pi^*$ in the constraint formulation Definition 4.1, we have

$$\mathbb{E}_{\rho_{\widehat{\pi}}^{M}} \left[ \log \rho_{\exp} - \log \rho_{\widehat{\pi}}^{M} \right] \geq \mathbb{E}_{\rho_{\pi^*}^{M}} \left[ \log \rho_{\exp} - \log \rho_{\pi^*}^{M} \right] \tag{26}$$

We then apply Lemma A.1 twice.

$$\mathbb{E}_{\rho_{\widehat{\pi}}^{M'}} \left[ \log \rho_{\exp} - \log \rho_{\widehat{\pi}}^{M'} \right] \geq \mathbb{E}_{\rho_{\pi^*}^{M'}} \left[ \log \rho_{\exp} - \log \rho_{\pi^*}^{M'} \right] - O(\frac{\gamma}{1-\gamma}\sqrt{\varepsilon^*}).$$

Finally, we have

$$\mathbb{D} \left( \rho_{\widehat{\pi}}^{M'} \parallel \rho_{\exp} \right) \leq \mathbb{D} \left( \rho_{\pi^*}^{M'} \parallel \rho_{\exp} \right) + O(\frac{\gamma}{1-\gamma}\sqrt{\varepsilon^*}).$$

$\square$

# B   Full Experiments

## B.1   Hyperparameters

To use the popular framework of Adversarial Imitation Learning, we use the regularization method and their hyperparameters according to previous papers (e.g. Brantley et al., 2019; Orsini et al., 2021). Besides the policy learning algorithm and the discriminator in AIL framework, our algorithm DYNAIL also includes two classifiers $q_{\text{sa}}$ and $q_{\text{sas}}$, and we use the same architecture and hyperparameter as the discriminator. For the parameter $\eta$, we explain our methods in the next subsection. The hyperparameters are shown in Table 1.

Table 1: Hyperparameters in Algorithm 1

| Hyperparameter | Values considered | Final Value |
|---|---|---|
| Parallel Environments | 8, 10 | 10 |
| $\ell_2$ regularization | 0 | 0 |
| Entropy coefficient | 0.01 | 0.01 |
| Gradient clipping | 0.1, 1.0 | 1.0 |
| Policy learning rate | $3 * 10^{-4}$ | $3 * 10^{-4}$ |
| Generator batchsize | 100, 200 | 100 |
| Discriminator learning rate | $1 * 10^{-3}$ | $1 * 10^{-3}$ |
| Discriminator batchsize | 800, 1000 | 1000 |
| Discriminator updates | 5, 10 | 5 |
| Discriminator weight decay | 0.01, 0.1 | 0.01 |
| Discriminator logits clipping | 5, 10, 20 | 20 |
| Classifier learning rate | $1 * 10^{-3}$ | $1 * 10^{-3}$ |
| Classifier batchsize | 800, 1000 | 1000 |
| Classifier updates | 5, 10 | 5 |
| Classifier weight decay | 0.01, 0.1 | 0.01 |
| Discrepancy Regularization $\Phi$ clipping | 5, 10, 20 | 5 |

## B.2 Ablation study on parameter $\eta$

The workhorse of Algorithm 1 is incorporating the dynamics variation into the state-action occupancy measure matching as a regularization term. When we optimize the policy via the reward function

$$\widetilde{r}(s, a, s') = \log(D_\theta) - \log(1 - D_\theta) + \eta \left( \log \frac{q_{\text{sas}}}{1 - q_{\text{sas}}} - \log \frac{q_{\text{sa}}}{1 - q_{\text{sa}}} \right),$$

it is important to select the tuning parameter $\eta$. Also, to achieve the $\varepsilon^*$ in Definition 4.1 and Theorem 4.1, we need to use an appropriate $\eta$. In this section, we describe our method to select this hyperparameter, and present the empirical performance.

Recall that our goal is to minimize the reverse KL divergence via Equation (7), which is

$$\min_\pi \mathbb{D}\left(p_\pi(\tau; \text{target}) \parallel p_\theta(\tau; \text{source})\right).$$

When we use Algorithm 1 with different hyperparameter $\eta$, we obtain the learned policy $\widehat{\pi}_\eta$. We use the "reverse KL divergence" between the $\widehat{\pi}_\eta$ and the expert demonstrations as a metric to select the hyperparameter $\eta$. More specifically, we train a classifier $Q_\eta$ for the $(s, a, s')$ between the expert demonstrations $\mathcal{D}_{\text{demo}}$ and $\widehat{\pi}_\eta$. Let $\mu$ denote the stationary distribution of $(s, a, s')$. Similar to the analysis of discriminator introduced in Section 4, the classifier $Q_\eta$ achieves its optimum at

$$Q_\eta(s, a, s') = \frac{\mu_{\text{exp}}(s, a, s')}{\mu_{\text{exp}}(s, a, s') + \mu_{\widehat{\pi}_\eta}(s, a, s')}.$$

Then the average of negative logit value on the generated trajectory equals

$$- \mathbb{E}_{(s,a,s') \sim \mu_{\widehat{\pi}_\eta}} \log \frac{\mu_{\text{exp}}(s, a, s')}{\mu_{\widehat{\pi}_\eta}(s, a, s')} = \mathbb{D}\left(\mu_{\widehat{\pi}_\eta} \parallel \mu_{\text{exp}}\right). \tag{27}$$

Hence we choose the hyperparameter $\eta$, which yields the smallest the $\mathbb{D}\left(\mu_{\widehat{\pi}_\eta} \parallel \mu_{\text{exp}}\right)$. We present the ablation study of $\eta$ and its corresponding performance for the target domain DisabledAnt-v1. in Figure 4. More specifically, we use the expert demonstrations from the source domain, apply different $\eta$ in $\{0.5, 1, 2, 4, 8 \cdots, 1024\}$, and run Algorithm 1 respectively. For different $\eta$, we calculate the Equation (27) for each $\widehat{\pi}_\eta$, and present it on the left plot of Figure 4. In the right plot of Figure 4, we present the normalized reward vs. iterations for different $\eta$. We observe that a smaller KL divergence for $\widehat{\pi}_\eta$ in the left plot leads to better performance of the normalized return in the right plot. Consistent to our theory, this provides an accurate method for model selection on different hyperparameter $\eta$.

## B.3 Experiments Details

In Section 5.2, we compare our algorithm with several baselines on different simulation platforms, which are previously studied in Eysenbach et al. (2021) and Dulac-Arnold et al. (2020). Details for these target environments are as follows, and the snapshots are provided in Figure 5.

**BrokenHalfCheetah-v3.** This environment is based on the source environment HalfCheetah-v3 from OpenAI Gym (Brockman et al., 2016). The goal of this environment is to make the agent move forwards. Episode length is 1000 steps long. This environment differs source environment from the 0th joint (0-indexed) which is broken: the input torque at this joint is ignored.

**HalfCheetahObstacle-v3.** This environment is based on the source domain HalfCheetah-v3 from OpenAI Gym (Brockman et al., 2016). The goal of this environment is to make the agent move forwards. Episode length is 1000 steps long. A key modification is that the agent is rewarded for running both forwards and backwards. Because it is easier to learn to run backwards, an obstacle behind the agent is used in this target environment. The agent bounces off the obstacle when running backwards and thus runs forwards for more reward.

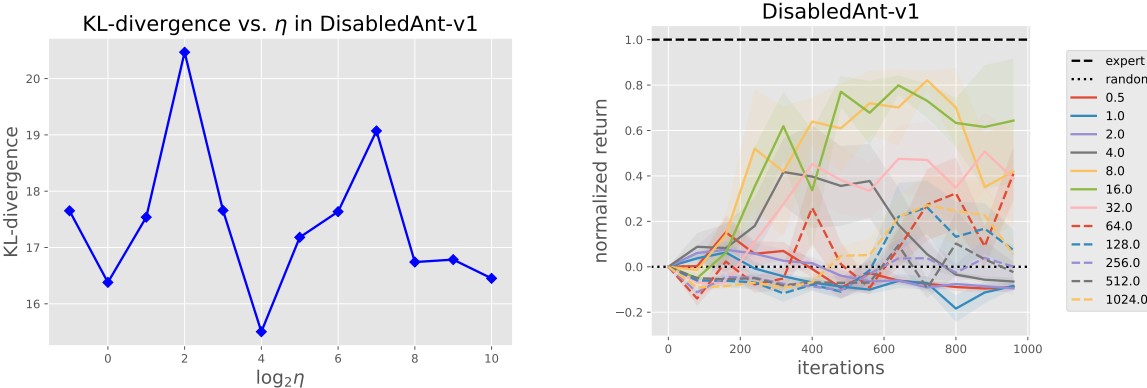

Figure 4: Ablation study results for the DisabledAnt-v1 environment. The left plot shows the $\mathbb{D}\left(\mu_{\widehat{\pi}_\eta} \parallel \mu_{\exp}\right)$ for different $\widehat{\pi}_\eta$. The right plot shows the normalized reward vs. iterations for different $\eta$. The shaded area stands for one standard deviation for 20 trials. We observe a smaller KL divergence for $\widehat{\pi}_\eta$ does lead to better performance of the normalized return. Consistent to our theory, this provides an accurate method for model selection on different hyperparameter $\eta$.

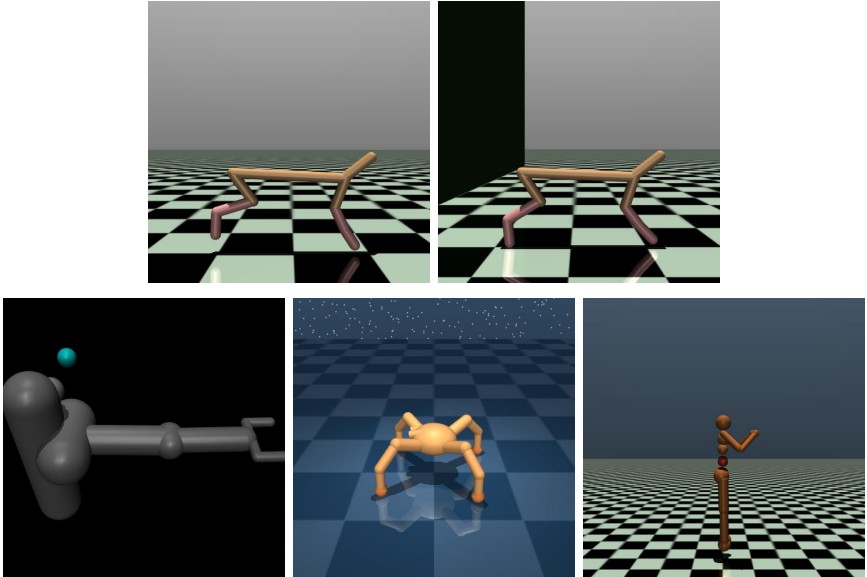

Figure 5: The snapshot in the corresponding *target environment* (BrokenHalfCheetah-v0, HalfCheetahObstacle-v0, BrokenReacher-v0, LowFrictionQuadruped and BrokenHumanoid-v3 are presented from left to right).

**BrokenReacher-v0.** The reacher in this environment is constructed by the 7DOF robot arm in Pusher environment from OpenAI Gym (Brockman et al., 2016). The goal of this environment is to move the end effector close to the target. Episode length is 100 steps long. This environment differs source environment from the 2nd joint (0-indexed) which is broken: the input torque at this joint is ignored.

**LowFrictionQuadruped.** This environment is based on the source domain "quadruped" with "real-walk" task from realworldrl-suite (Dulac-Arnold et al., 2020). The goal of this environment is to make the agent walk forwards. Episode length is 1000 steps long. The contact friction of the *source domain* ranges from 0.1 to 4.5 with the standard deviation of 0.5, while we lower the friction to 0.1 as a constant in the *target domain*. With a low contact friction, it is harder for the agent to walk forwards.

**BrokenHumanoid-v3.** This environment is based on the source environment Humanoid-v3 from OpenAI Gym (Brockman et al., 2016). The goal of this environment is to make the agent move forwards. Maximum episode length is 1000 steps long, however, episode may terminate early due to unhealthy conditions. This environment differs source environment from the 0th joint (0-indexed) which stands for the red broken abdomen in Figure 5: the input torque at this joint is ignored.

## C   Societal Impact

Real world Reinforcement Learning and Imitation Learning are closely related to our work (Dulac-Arnold et al., 2020; Kirk et al., 2021). Recent years witness the success of using RL and IL in a series of artificial domains. Nevertheless, many state-of-the-art Imitation Learning and Inverse RL algorithms are hard to deploy in real-world scenarios since the underlying assumptions are rarely satisfied in practice (Dulac-Arnold et al., 2020). Our research is aligned with previous papers – we study the generalization performance for Imitation Learning algorithms (more specifically, under dynamics variation). We mainly focus on the algorithm design and do not consider specific applications.

