# OpenReview forum: "Dynamics Adapted Imitation Learning"
_TMLR — Accepted by TMLR_

### Review · Reviewer_PYce · 2023-04-14

**Summary Of Contributions:**

The authors are interested in Imitation Learning with a dynamic shift between the demonstrations (given in a source domain) and the actual environment (the target domain).
The proposed algorithm (DYNAIL), builds upon the adversarial imitation learning (AIL) framework. The authors propose a modified version of the divergence minimization objective that takes into account the dynamics shift. They then propose an algorithm that requires training two other networks and is a modification of the “inverse RL reward” classically used in AIL using the confusion of the discriminator.
They propose theoretical guarantees and experiments validating their algorithms on a set of simulated tasks for MuJoCo.


**Audience:**

Yes

**Broader Impact Concerns:**

None.

**Claims And Evidence:**

Yes

**Requested Changes:**

I would like to see:

- more detailed justification of the assumption and objective choice.
experiments actually breaking the assumption and highlighting if DYNAIL still works or fails.
- an experiment in the lower data regime (see [1], or [2], for which higher dimensional tasks like Humanoid are successful with very little trajectories that are subsamples 20 times). I am not saying the subsampled setup is the most meaningful, but then what happens if you only have a handful of demonstrations?
- for the reason previously mentioned, I would have liked to see one experiment with a higher -dimensional action space like a humanoid  OR one that requires navigation (like AntMaze) with dynamic shift as presented in figure 1.
- for all experiments, please don’t use “iterations” for the x-axis which is not meaningful at all in RL. Please rather use the number of environment steps, as you are working with online imitation learning.
- Ideally, I would like to see results in terms of divergence too, as optimizing the reward is only a proxy to performance, as suggested in [2] or [3].


[1] What Matters for Adversarial Imitation Learning?

[2] Primal Wasserstein Imitation Learning

[3] Hyperparameter Selection in Imitation Learning


**Strengths And Weaknesses:**

**Strengths**

The paper is well written and correctly placed with respect to the literature. It is easy to follow and the main idea is simple, neat and well explained.


**Weaknesses**

On the fundamental part, I see one main weakness which is the following: I think it would be worth discussing much more the chosen objective (Eq.7). I know the reverse KL objective is widely used but in this specific case, I had to think quite a lot to try to make sure the objective made sense. This is also the case for Assumption 1, which I believe is much stronger than what the authors say in the paper. One extremely simple example breaking this assumption can be derived directly from the example present in Fig.1: what if, instead of “adding a wall”, we “move a wall”? The assumption is broken.

On the experimental part, I thus miss experiments that actually break this assumption. What happens in this case? Does DYNAIL still work in practice or does it fail? It would be a strength of the paper to highlight failure modes of the proposed algorithm.

---

> ### Author Response · Authors · 2023-05-16
> **Reply to Reviewer PYce**
>
> We sincerely value and appreciate your insightful feedback. In response to the issues you highlighted, we have offered further insights and conducted additional experiments to address these concerns. The newly incorporated details and experimental results can be located in our revised manuscript the anonymous repository (https://anonymous.4open.science/r/DYNAIL-4039).
>
> 1. **Q: Experiments actually breaking the assumption and highlighting if DYNAIL still works or fails.**
>
>     As addressed in **Appendix B.1**, Assumption 1 is due to the mathematical definition of the widely used Reverse KL divergence. Techniques such as clipping the logits of the model (which is equivalent to adding a small constant to the probabilities to ensure that they are never exactly zero) can alleviate the issue.
>
>     We also would like to point out that our assumption can be regarded as a light-weight assumption in the field of domain adaptation in IL compared with prior works. Assumptions that have access to online expert (Kim et al., 2020) and (multiple) source domains (Chae et al., 2022) are much stronger.
>
>     Moreover, if Assumption 1 does not hold, then the optimal policy for the target domain might involve behaviors that are not possible in the source domain, so it is unclear how one could learn a near-optimal policy by learning from the demonstrations in the source domain.
>
>     The experimental results provide a compelling justification for our method. We already have a couple of domains, HalfCheetah-v3 and HalfCheetahObstacle-v3, that break the assumption to some extent, which is also discussed in Eysenbach et al. (2021). HalfCheetahObstacle-v3 does not entirely satisfy the assumption because transitions such as **bouncing off the obstacle are only possible in the target domain, not the source domain**. As you can see in the **Figure 3**, DYNAIL still works.
>
>     In reponse to your conerns, we add maze experiments to test the performance of DYNAIL when the assumption is not satisfied. We modified the wall in the source domain by moving the middle block to the right. The GIF is uploaded to the **anonymous repository (https://anonymous.4open.science/r/DYNAIL-4039)**. In the experiments, **DYNAIL can find the near-optimal trajectory which does not even exit in the source domain**. Therefore, the success of our method on this task illustrates that DYNAIL can excel even in settings that do not satisfy Assumption 1.
>
> 2. **Q: What happens if you only have a handful of demonstrations.**
>
>     Although our method is not designed for the minimum amount of expert demonstrations, we present the performance if we reduce the demonstrations from 40 to 1 (40-DYNAIL and 1-DYNAIL). We also take SOTA method GWIL (40-GWIL and 1-GWIL) for comparison. It should be noted that we extend 1-GWIL (GWIL using the top one of 40 demonstrations) to 40-GWIL by sampling from 40 demonstrations to compute the mean Gromov-Wasserstein reward.
>
>     |  Environments  | 40-DYNAIL  |  1-DYNAIL   | 40-GWIL  |  1-GWIL   |
>     |  :----:  | :----:  |  :----:  | :----:  | :----:  |
>     | CustomAnt-v0 | **0.93** | 0.80 | 0.82 | **0.91** |
>     | DisabledAnt-v1 | **0.80** | 0.26 | 0.48 | 0.52 |
>     | DisabledAnt-v2 | **0.63** | 0.43 | 0.29 | 0.32 |
>     | BrokenHalfCheetah-v3 | **0.31** | 0.15 | 0.02 | 0.20 |
>     | BrokenHumanoid-v3 | **0.84** | **0.83** | 0.04 | 0.05 |
>     | BrokenReacher-v0 | **0.90** | 0.67 | 0.19 | 0.35 |
>     | LowFrictionQuadruped | **0.68** | **0.68** | 0.53 | **0.68** |
>     | HalfCheetahObstacle-v3 | **1.44** | 1.37 | 1.01 | 1.20 |
>
>     We notice that the performance of DYNAIL decreases a lot in the disabled ant environments and BrokenHalfCheetah-v3 as expected. Surprisingly, we can also see that in other environments, 1-DYNAIL achieves a competitive performance which we attribute to the matching degree of the only demonstration and target domain. By the way, 1-GWIL always outperforms 40-GWIL for it is originally designed for one demonstration. Therefore we always present the performance of 1-GWIL to compare with DYNAIL.

---

> ### Author Response · Authors · 2023-05-16
> **Reply to Reviewer PYce**
>
> 3. **Q: I would have liked to see one experiment with a higher -dimensional action space like a humanoid.**
>
>     As is shown in **Figure 3**, we now add experiments of BrokenHumanoid-v3 (humanoid with broken abdomen) to test the performance of DYNAIL under high-dimensional environment
>
>     |  Methods  | BrokenHumanoid-v3   |
>     |  :----:  | :----:  |
>     | BC | 0.03 |
>     | AIRL | -0.02 |
>     | SOAIRL | -0.01 |
>     | EAIRL | -0.01 |
>     | DYNAIL | **0.84** |
>     | GWIL | 0.05 |
>     | GAIFO | **0.80** |
>
>     From the table, we can see that DYNAIL can obtain a near-optimal policy in humanoid task. We note that we change the RL algorithm in Algorithm 1 from PPO to SAC, because humanoid task is too hard for PPO to even get an expert policy, which means PPO does not have the ability to learn from a ground-truth reward function. When we use SAC to serve as the generator, DYNAIL achieves a good performance. More details for the humanoid experiments can be seen in the **anonymous repository (https://anonymous.4open.science/r/DYNAIL-4039).**
>
> 4. **Q: Please rather use the number of environment steps, as you are working with online imitation learning.**
>
>     Thanks for your valuable advice, we now change all the x-axis to environment steps. It is direct to make the change because **we explore 1000 steps before each iteration** in all the experiments. Therefore, it is obvious that 1000 iterations stands for 1e6 steps in all the figures. The changes can be seen in our **revised manuscript**.
>
> 5. **Q: I would like to see results in terms of divergence too.**
>
>     We completely understand that you are concerned about the direct objective of our method. However, it is difficult to obtain the divergence of trajectory in Equation (7). We equally maximize the return with discrepancy regularization (Equation (9)) to replace the minimization of divergence (Equation (7)), so we can not directly obtain the divergence as the Wasserstein distance in [2]. It is also infeasible for us to compute the divergence as the state divergence in [3]. Because the dynamics of source and target domain are the same in [3], the dynamics can be reduced when computing the state divergence. However, when there is dynamics variation, things get totally different. We have no access to the dynamics of environments and thus can not compute the divergence directly. In order to give you intuitive feeling of divergence, we use the technique mentioned in Appendix C.2 to approximate the divergence of the final policy.
>
>     |  Environments  | DYNAIL  |  GWIL   |
>     |  :----:  | :----:  |  :----:  |
>     | CustomAnt-v0 | **1.23** | 8.53 |
>     | DisabledAnt-v1 | **11.35** | 13.39 |
>     | DisabledAnt-v2 | **10.56** | 14.59 |
>     | BrokenHalfCheetah-v3 | **8.99** | 12.77 |
>     | BrokenHumanoid-v3 | **16.95** | 20.83 |
>     | BrokenReacher-v0 | **9.41** | 12.58 |
>     | LowFrictionQuadruped | **13.28** | 24.56 |
>     | HalfCheetahObstacle-v3 | **3.97** | 8.22 |
>
>     We can see that KL-divergence of DYNAIL is always lower than GWIL, which corresponds to the objective of our method.

---

> > ### Comment · Reviewer_PYce · 2023-05-17
> > **Thanks for the additional work.**
> >
> > Thanks for the clear explanations. The additional experiments are convincing and, in my opinion, fill an important gap in the paper's story (notably the question of what happens when "breaking the assumption 1"). The broken humanoid results are also strong and valuable.
> >
> > This is a large amount of work produced by the authors and I would like to thank them for this. I also read the answers to other reviewers and find that they tackle the raised concern with clear answers.

---

> > > ### Author Response · Authors · 2023-05-19
> > > **Thanks for your positive feedback.**
> > >
> > > We appreciate your acknowledgement of the hard work and dedication we have put into our research. We strive to conduct thorough experiments and consider every possible scenario, including the instance of "breaking Assumption 1". The broken humanoid results were an important aspect of our study and we're glad you found them valuable.
> > >
> > > It was our responsibility to respond comprehensively to the reviewers' concerns, and we're pleased that you find our responses satisfactory. Thank you again for recognizing our efforts and providing positive feedback.

---

### Review · Reviewer_tc73 · 2023-04-17

**Summary Of Contributions:**

The paper tackles the problem of imitation learning when the expert demonstrations were collected under different system dynamics. The paper proposes a new method, DYNAIL, to better imitate the expert demonstrations despite the mismatch of the system dynamics. DYNAIL uses adversarial imitation learning to minimize the reverse Kullback-Leibler divergence, similar to AIRL (However, DYNAIL does not seem to use the special form of discriminator used by AIRL). Different from previous adversarial imitation learning methods, DYNAIL introduces an additional cost given by the expected KL-divergence $D(p(s'|s,a) || p'(s'|s,a))$ between the transition dynamics on the target domain and those on the source/expert domain. This expected KL divergence is estimated using two additional discriminators, one trained on the state transitions (s',s,a) and the second one on the system inputs (s,a) [A similar procedure was proposed by EAIRL]. The paper also provides a bound on the imitation gap on the *source* domain. The approach is evaluated on Mujoco-Ant where the dynamics by changing the kinematics (e.g. leg length) of the robot. Compared to related work, such as EAIRL and GWIL that also tackle imitation learning under dynamic mismatch, the proposed method achieves better final performance.

**Audience:**

Yes

**Claims And Evidence:**

No

**Requested Changes:**

The claim that "Expanding equation (7) yields the policy objective Equation (8)" needs to be substantiated. The additional cost term should be derived from the distribution matching objective.

Ideally, the theoretical analysis should be extended to the *target* domain. The limitations of the current analysis should be clearly stated.

**Strengths And Weaknesses:**

Strength:
- The description of the method is sufficiently clear
- The paper tackles an important problem
- The method shows reasonable performance in the problem setting
- penalizing the dynamics mismatch using the reverse KL is sound, as it will bias the agent to state-action pairs that produce similar transitions compared to the source domain.

Weaknesses:
- As far as I can tell, the additional term of the reward function was not derived but heuristically chosen. Although the paper states that
"Expanding equation (7) yields the policy objective Equation (8)", this derivation is not shown in the paper, Furthermore, it seems quite clear that Equation (8) can not directly be derived from the distribution matching objective (Eq. 7), as it contains an additional hyperparameter $\eta$ for weighting the cost for the dynamics mismatch. The approach is only justified by showing that the price (imitation gap) of this additional cost term can be bounded.

- The theoretical analysis is restricted to the source domain. However, on the source domain, prior imitation learning methods (that do not use the addition cost for biasing the policy) enjoy better theoretical guarantees. Instead, it would be much useful to bound the imitation gap on the *target* domain.

---

> ### Author Response · Authors · 2023-05-16
> **Reply tp Reviewer tc73**
>
> We sincerely value and appreciate your insightful feedback. In response to the issues you highlighted, we have offered further insights and conducted additional experiments to address these concerns. The newly incorporated details and experimental results can be located in our revised manuscript and the anonymous repository (https://anonymous.4open.science/r/DYNAIL-4039).
>
> 1. **Q: The claim that "Expanding equation (7) yields the policy objective Equation (8)" needs to be substantiated.**
>
>     Equation (7) is as follows:
>
>     $$
>     \min_\pi \mathbb{D} \left({p_\pi(\tau; {\rm{target}})} \mid\mid {p_\theta(\tau; {\rm{source}})}\right)
>     $$
>
>     With Equation (5) and (6) in hand, Equation (7) can be derived as:
>
>     $$
>     \\begin{align*}
>     \min_\pi \mathbb{D} \left({p_\pi(\tau; {\rm{target}})}\mid\mid{p_\theta(\tau; {\rm{source}})}\right) &= \min_\pi \sum_\tau{p_\pi(\tau;
>     {\rm{target}})}\log \frac{p_\pi(\tau; {\rm{target}})}{p_\theta(\tau; {\rm{source}})} \\\\
>     &= \min_\pi \mathbb{E}\_{\tau \sim \pi, M} \Big[\log \frac{p(s_0) \prod\_{t} M(s\_{t+1}\mid  s_t, a_t) \pi(a_t\mid  s_t)}{\frac{1}{Z} p(s_0)
>     \times \prod\_{t} M'(s\_{t+1}\mid  s_t, a_t) \exp( r_\theta(s_t, a_t) /\eta)} \Big] \\\\
>     &= -\max_\pi \mathbb{E}\_{\tau \sim \pi, M} \Big[\sum_t r_\theta(s_t, a_t)/\eta - \log \pi(a_t \mid s_t) + \log \frac{M'(s\_{t+1}\mid  s_t, a_t)}
>     {M(s\_{t+1}\mid  s_t, a_t)} - \log Z\Big] \\\\
>     &\propto -\max_\pi \mathbb{E}\_{\tau \sim \pi, M} \Big[\sum_t r_\theta(s_t, a_t) - \eta \log \pi(a_t \mid s_t) + \eta \log \frac{M'(s\_{t+1}\mid
>     s_t, a_t)}{M(s\_{t+1}\mid  s_t, a_t)} \Big]
>     \\end{align*}
>     $$
>
>     It is noted that we leave out $\log Z$ as the constant in the last line. Then, we use ${\Phi}(s_t, a_t, s_{t+1})$ as the notation of the discrepancy regularization and obtain Equation (8):
>
>     $$
>     \max_{\pi } \mathbb{E}\_{\tau \sim \pi, M}
>         \Big[ \sum_t  r_\theta(s_t, a_t) - \eta \log \pi(a_t \mid s_t) + \underbrace{\eta {\Phi}(s_t, a_t, s_{t+1})}\_{\text{Discrepancy Regularization}}
>         \Big]
>     $$
>
> 2. **Q: The limitations of the current analysis should be clearly stated.**
>
>     We concur with your observation about the limitation of our analysis to the source domain, and agree that extending the bounds to the target domain would enhance our work's value. This limitation has been more explicitly stated in our revised manuscript (**see Remark 4.1**).
>
>     Due to the nature of Imitation Learning, we compare $\hat{\pi}$ (the output of our method) to the $\pi_{\exp}$ (provided expert demonstration), and our Theorem 4.1 naturally integrates $\pi_{\exp}$ in the source domain and $\pi^*$, which is the optimal policy in target domain. While prior methods can achieve nearly zero on the LHS of Theorem 4.1, they lack any mechanism for adapting to dynamic changes — a critical aspect our method addresses.
>
>     We believe that our theorem and experimental results together provide a compelling justification for our method; We appreciate your constructive feedback to enhance our future research.

---

> > ### Comment · Reviewer_tc73 · 2023-05-24
> > **Thank you for the clarifications**
> >
> > Thank you for the reply. However, in your derivation you express $p_\theta(\tau)$ in terms of $r_\theta$. While this relation makes sense in the original MaxEnt-IRL formulation by Ziebart, for adversarial imitation learning, the exponentiated discriminator reward will not be proportional to the expert likelihood. While I agree that the GAIL-style alternation between policy-updates and reward-updates may in practice also lead to approximately matching the expert distribution, even when adding the additional bias towards transitions where the dynamics of source and target domain are similar, I still do not find a theoretical justification that DYNAIL minimizes the KL-divergence to the source-domain expert distribution.
> >
> > Also, the origin of $\eta$ is still not fully clear to me.  Eq. (1) claims that $\eta$ is a temperature parameter of MaxEnt-IRL. However, MaxEnt-IRL does not involve a temperature parameter.

---

> > > ### Author Response · Authors · 2023-05-24
> > > **Thank you for the reply.**
> > >
> > > Thanks for your reply. We have endeavored to respond to each concern you raised.
> > >
> > > 1. **Q: The relationship between the exponentiated reward function and the expert likelihood.**
> > >
> > >     As stated in AIRL (Fu et al., 2018), updating the discriminator can be viewed as updating the reward function, and updating the policy can be viewed as improving the sampling distribution used to estimate the partition function. If trained to optimality, the exponential optimal reward function will match the expert likelihood.
> > >
> > >     The Equation (1) connects the reward learning problem with maximum likelihood problem,
> > >     $$
> > >     \max_\theta \mathbb{E}\_{\tau \sim \mathcal{D}\_{\rm{demo}}} [\log p_\theta(\tau)], \text{ where } p_\theta(\tau) = \frac{1}{Z} p(s_0) \times \prod\_{t=0}^T M(s\_{t+1}\mid  s_t, a_t) \exp(r_\theta(s_t, a_t)/\eta).
> > >     $$
> > >     The justification of minimizing KL-divergence for adversarial imitation learning can also be found in AIRL (**Fu et al., 2018, Appendix A**)
> > >
> > > 2. **Q: The origin of temperature parameter.**
> > >
> > >     The temperature parameter $\eta$ is considered a common modification to the framework of RL and probabilistic inference (**Levine, 2018 Section 2.5**), and is widely used in prior works (Liu et al., 2022a).
> > >
> > >
> > > Justin Fu, Katie Luo, and Sergey Levine. Learning robust rewards with adverserial inverse reinforcement learning. In International Conference on Learning Representations, 2018.
> > >
> > > Sergey Levine. Reinforcement learning and control as probabilistic inference: Tutorial and review[J]. arXiv preprint arXiv:1805.00909, 2018.
> > >
> > > Jinxin Liu, Zhang Hongyin, and Donglin Wang. DARA: Dynamics-aware reward augmentation in offline reinforcement learning. In International Conference on Learning Representations, 2022a.

---

> > > > ### Comment · Reviewer_tc73 · 2023-05-24
> > > > **I need further clarifications**
> > > >
> > > > AIRL uses a special form of the discriminator that subtracts $\log \pi(a,s)$ to the computed logits, Do you also use this form of discriminator in your approach? I can not find that in the manuscript.
> > > >
> > > > Equation 1 refers to MaxEnt-IRL (Ziebart 2008, 2010). This equation (without temperature) is derived based on the optimization problem of maximizing the entropy subject to a feature matching constraint.  What optimization problem was used for deriving Eq. 1 (which includes the temperature parameter)?  The provided references consider the MaxEnt reinforcement learning setting, where a temperature can be straightforwardly included.

---

> > > > > ### Author Response · Authors · 2023-05-25
> > > > > **Further Clarifications**
> > > > >
> > > > > 1. **Q: The form of discriminator.**
> > > > >
> > > > >     We actually use a similar form with AIRL's discriminator, $D_\theta = \frac{\exp(r_\theta/\eta-\Phi)}{\log \pi + \exp(r_\theta/\eta-\Phi)}$, which subtracts both the entropy term and the discrepancy regularization term to the computed logits, thus leading to a similar jusification.
> > > > >
> > > > >     The form of DYNAIL's discriminator can be derived from Equation (9) in the manuscript,
> > > > >     $$
> > > > >     \widetilde{r}(s, a, s') = \log(D_\theta) - \log(1-D_\theta) + \eta \left( \log\frac{q_{\rm sas}}{1-q_{\rm sas}} - \log\frac{q_{\rm sa}}{1-q_{\rm sa}} \right).
> > > > >     $$
> > > > >     And we agree to add the explicit DYNAIL's discriminator form to the manuscript to help understand the justification.
> > > > >
> > > > > 2. **Q: The parameter $\eta$.**
> > > > >
> > > > >     Firstly, we would like to state that DYNAIL is a method based on Max-Ent IRL. Max-Ent IRL seeks to infer the reward function $r_\theta(s, a)$ given a set of expert demonstrations. In IRL, we assume the demonstrations are drawn from an optimal policy and the optimization problem can be interpreted as solving the maximum likelihood problem as Equation (1) without the parameter $\eta$ (**Fu et al., 2018, Section 3 & Appendix A**).
> > > > >
> > > > >     Secondly, adding a scale parameter $\eta$ does not actually change the generality of the maximum likelihood problem. Besides, the scale parameter $\eta$ is commonly used in prior IRL works (for example, **Scobee & Sastry, 2020, Section 3.3**), $\eta \in [0, \infty)$ is a parameter describing how closely an agent adheres to the task of optimizing the reward function (as $\eta \rightarrow 0$, the agent becomes a perfect optimizer, and as $\eta \rightarrow \infty$, the agent’s actions become perfectly random).
> > > > >
> > > > > Justin Fu, Katie Luo, and Sergey Levine. Learning robust rewards with adverserial inverse reinforcement learning. In International Conference on Learning Representations, 2018.
> > > > >
> > > > > DRR Scobee, SS Sastry. Maximum Likelihood Constraint Inference for Inverse Reinforcement Learning. In International Conference on Learning Representations, 2020

---

### Review · Reviewer_hbCj · 2023-05-11

**Summary Of Contributions:**

This paper proposes DYNAIL - an adversarial imitation method that accounts for changes in transition dynamics from the source to the target environment. The paper proposes a practical algorithm that consists of the discrepancy regularizer to compensate for the dynamics model difference. The paper also establishes an upper bound for divergence from expert policy which depends on transition dynamics differences between source and target. The paper showed empirical validation on multiple continuous domain tasks showing that DYNAIL outperforms previous baselines.

**Audience:**

Yes

**Broader Impact Concerns:**

No major ethical concern

**Claims And Evidence:**

Yes

**Requested Changes:**

1. Could you please discuss how the proposed approach compares to Imitation Learning from Observation kind of methods like GAIfO and BCO mentioned above? These methods learn directly from the observations in the demonstration that can transfer to target environments with different dynamics.

2. Why was this method not applied to any discrete action domain tasks? For example, a FlappyBird environment with higher/lower gravity in the target setting can be a good experiment or something else with discrete actions.


**Strengths And Weaknesses:**

Strength:
The paper shows good empirical evidence showing that the proposed method performs well compared to previous baselines. The DYNAIL model proposes to close the gap between different environment dynamics between train and testing by using


Weakness:
However, this paper does not provide experimental validation on discrete domain environments. Superior performance on such environments would make the claim of the paper stronger and more general. Additionally, the problem of adapting to target environments with different dynamics can be also handled by imitation learning from observation methods like Behavior Cloning from Observations (BCO)[1] and Generative Adversarial Imitation from Observations (GAIfO)[2]. Since these methods learn from observations, they can handle dynamics model differences between source and target. Furthermore, some previous works (such as [3]) also attempts adversarial imitation learning for different target environments. Discussion related to the differences from these works might be useful.

[1] Torabi, F., Warnell, G. and Stone, P., 2018, July. Behavioral cloning from observation. In Proceedings of the 27th International Joint Conference on Artificial Intelligence (pp. 4950-4957).

[2] Torabi, F., Warnell, G. and Stone, P., 2018. Generative adversarial imitation from observation. arXiv preprint arXiv:1807.06158.

[3] Desai, S., Durugkar, I., Karnan, H., Warnell, G., Hanna, J. and Stone, P., 2020. An imitation from observation approach to transfer learning with dynamics mismatch. Advances in Neural Information Processing Systems, 33, pp.3917-3929.

---

> ### Author Response · Authors · 2023-05-16
>
> We sincerely value and appreciate your insightful feedback. In response to the issues you highlighted, we have offered further insights and conducted additional experiments to address these concerns. The newly incorporated details and experimental results can be located in our revised manuscript and the anonymous repository (https://anonymous.4open.science/r/DYNAIL-4039).
>
> 1. **Q: How the proposed approach compares to Imitation Learning from Observation kind of methods like GAIfO and BCO mentioned above?**
>
>     We do pay attention to IfO methods such as Liu et al. (2020) discussed in the related work. Also, we already have state-only version of AIRL (SOAIRL) to the baselines to make comparisons with our method. Similar with SOAIRL, BCO and GAIFO can be respectively regarded as a state-only version of BC and GAIL with stronger ability to transfer policy. Since GAIFO has better performance than  BCO, **we add GAIFO to our baselines in the revised manuscript** and the results shows that DYNAIL outperforms SOAIRL and GAIFO. The results are reasonable for IfO methods because they dismiss the action information in the demonstrations which they think might be misleading. In the contrast, our method leverages this information and achives better performance.
>
> 2. **Q: Why was this method not applied to any discrete action domain tasks?**
>
>     Thanks for your and reviewer PYce’s advice, we now add maze experiments related to Figure 1 to better illustrate our method. It is clear that maze domains with four discrete actions in the action space are discrete domains. Meanwhile, they can test the performance of DYNAIL when Assumption 1 is not satisfied. It is uninformative to show the reward curve of maze experiments for sparse reward (either 0 or final reward). Therefore, we upload the GIF of maze experiments to the **anonymous repository (https://anonymous.4open.science/r/DYNAIL-4039)** to help understand how DYNAIL works. In the experiments, **DYNAIL can find the near-optimal trajectory which does not even exit in the source domain**. It is clear that DYNAIL can still work in the discrete domains where even Assumption 1 is not satisfied.

---

### Author Response · Authors · 2023-05-23
**Deadline for discussion approaching**

Dear Reviewers,

As the end of the discussion is approaching, we would like to express our gratitude once more for your valuable insights and feedback. We have made every effort to clarify the theoretical contribution, enhance the experimental section and demonstrate the effectiveness of proposed method DYNAIL. If you have any remaining questions regarding the contributions of this paper, please do not hesitate to ask. We are more than happy to engage in further discussions to address any comments, concerns, or questions you may have. Thanks again.

---

### Author Response · Authors · 2023-05-31
**The summary of main changes in the latest revision and additional details.**

Dear Reviewers,

We are extremely grateful for your valuable feedback and insightful comments. Your concrete suggestions are a valuable step in this direction, and we have revised our submission accordingly to take these into account. In this short note we summarize the main changes in the latest revision of our submission. We have also included the additional experimental results and details in the anonymous repository (https://anonymous.4open.science/r/DYNAIL-4039).

1. **Section 3.2**: Add the explicit form of discriminator to **paragraph 'Training the discriminator'**.
2. **Section 4**: Add more explicit limitation of Theorem 4.1 to **Remark 4.1**.
3. **Section 5**: Add a broken humanoid task to the experiments in **Figure 3**.
4. **Section 5**: Add a new baseline GAIFO to all the experiments in **Figure 2&3**.
5. **Section 5**: Change all the x-axis from 'iterations' to 'steps' in **Figure 2&3**.
6. **Appendix C.3**: Add a detailed description of the broken humanoid task.
7. Additional maze experiments are presented through the GIFs in the repository.
8. Additional experiments with less demonstrations are presented through the table in the response and the figure in the repository.
9. Additional divergence results are presented through the table in the response and the figure in the repository.

---

### Decision · Action_Editors · 2023-06-25

**Recommendation:** Accept as is

**Comment:**

The paper studies the important problem of imitation learning when source and target environments differ in dynamics. Reviewers agree the method is solid, and empirical evaluation is convincing (after adding new ones in the revised version). The theoretical justification is limited, but the revised version added an explicit discussion about the limitations. Overall, the work is an interesting and well-supported contribution.

**Audience:**

Imitation learning and dealing with source-target domain differences are both important topics. This work is expected to be interesting to a broad range of audience.

**Claims And Evidence:**

The paper studies imitation learning in the presence of dynamic differences between source and target environments. It proposes a new method (DYNAIL), inspired by AIRL, which is able to effectively imitate experts while dealing with system dynamic differences. The authors provided some form of theoretical justifications (although somewhat limited), and also demonstrate the performance empirically on benchmarks.